

Atmospheric
Measurement
Techniques

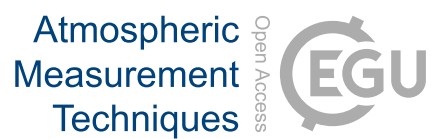

# Prediction model for diffuser-induced spectral features in imaging spectrometers

**Florian Richter**[1,2], **Corneli Keim**[2], **Jérôme Caron**[TS1][a], **Jasper Krauser**[2], **Dennis Weise**[2], **and Mark Wenig**[1]

[1][CE1]Meteorological Institute, Ludwig Maximilian University of Munich, Munich,
Theresienstraße 37, 80333 Munich, Germany
[2]Airbus Defence and Space GmbH, Willy-Messerschmitt-Straße 1, 82024 Taufkirchen, Germany
[a]now at: Optics Department, Netherlands Organization for Applied Scientific Research (TNO),
Stieltjesweg 1, 2628 CK Delft, the Netherlands

**Correspondence:** Florian Richter (flo.richter@physik.lmu.de)

**Abstract.** [TS2]Wide-field spectrometers for Earth observation missions require in-flight radiometric calibration for which the Sun can be used as a known reference. Therefore, a diffuser is placed in front of the spectrometer in order to scatter the incoming light into the entrance slit and provide homogeneous illumination. The diffuser, however, introduces interference patterns known as speckles into the system, yielding potentially significant intensity variations at the detector plane, called spectral features.

There have been several approaches implemented to characterize the spectral features of a spectrometer, e.g., end-to-end measurements with representative instruments. Additionally, in previous publications a measurement technique was proposed, which is based on the acquisition of monochromatic speckles in the entrance slit following a numerical propagation through the disperser to the detection plane. Based on this measurement technique, we present a stand-alone prediction model for the magnitude of spectral features in imaging spectrometers, requiring only few input parameters and, therefore, mitigating the need for expensive measurement campaigns.

## 1 Introduction

Many current and future Earth observation missions carry wide-field spectrometer payloads, such as the Envisat Medium Resolution Imaging Spectrometer (Olij et al., 1997), the Sentinel-2 Multispectral Imager (Martimort et al., 2012), the Sentinel-3a Ocean and Land Colour Imager (Nieke and Mavrocordatos, 2017), the Sentinel-4/UVN instrument (Clermont et al., 2019), the Sentinel-5/UVNS instrument (Guehne et al., 2017), and the Greenhouse Gas Information System (GHGIS) instrument of CO2M or the former CarbonSat (Fletcher et al., 2015). These space-based instruments require in-flight radiometric calibration for which the Sun can be used as a known reference. In order to ensure homogeneous illumination of the instrument, a diffuser is used to scatter the incoming sunlight into the entrance slit. However, the diffuser introduces an interference pattern known as speckles into the optical system. The speckles propagate through the disperser and are integrated at the detector plane, yielding intensity variations described as *spectral features*[CE2] by van Brug et al. (2004). When the solar spectrum is used for the calibration of the top of atmosphere measured spectra, the spectral features create a radiometric error as they alter the calibration function. Speckle effects are commonly known from applications involving highly coherent laser light, such as holographic imaging (Bianco et al., 2018) or laser speckle contrast imaging (Heeman et al., 2019). In general, diffuse sunlight does not yield a significant net speckle pattern when incident on a detector due to its broad spectrum. However, for spectrometers with a fine spectral resolution, the spectral width per channel is narrow enough to give rise to an amplitude of the spectral features that can be as large as other radiometric errors, e.g., due to straylight and polarization. Spectral features depend on various geometric conditions, implying that their exact position at the detec-

tor plane cannot be reliably predicted. This also renders any mitigating post-processing steps ineffective. Widely known speckle suppression techniques, such as the rotation or tilting of elements in the optical system (Goodman, 2007, Sect. 5), are only viable for on-ground calibration with a static setup. For space applications, additional moving parts are typically not implemented because they pose a supplementary risk of failure. In early planning phases, each contribution to the radiometric error needs to be quantitatively estimated to allow a global optimization of the instrument. This work presents a novel method to predict the radiometric error due to spectral features. As in most applications, the radiometric error is driven by the amplitude of the spectral features, and van Brug and Courrèges-Lacoste (2007) introduced the spectral features amplitude (SFA) as a standardized measure.

The issue of diffuser-induced spectral features in imaging spectrometers was first documented by Richter et al. (2002) and Wenig et al. (2004) in the context of the Global Ozone Monitoring Experiment (GOME). Ahlers et al. (2004) and van Brug et al. (2004) observed spectral oscillations caused by the onboard diffuser in the Scanning Imaging Absorption Spectrometer for Atmospheric Chartography (SCIA-MACHY) instrument. Several approaches to characterization and modeling have been proposed since. van Brug and Courrèges-Lacoste (2007) presented an end-to-end measurement setup, featuring a complete instrument, including a source, telescope optics, disperser, and detector. However, since spectral features need to be suppressed by the design of certain instrument parameters, a quantitative analysis with a representative spectrometer is usually difficult to achieve. van Brug and Scalia (2012) introduced models for different speckle averaging effects. However, a comprehensive and reliable model has not been presented yet. Isolating the spectral features by eliminating all other error sources in a representative end-to-end setup remains the main challenge with respect to gaining quantitative insights into the SFA dependence.

A different approach to quantifying spectral features was taken by Burns et al. (2017) and improved by Richter et al. (2018). It is based on the subsequent acquisition of monochromatic speckle patterns in the slit plane over several spectral channels, which are then propagated numerically through the disperser to the detection plane. Some simplifying assumptions are made about the optical system, which reduces the complexity and limits systematic error contributions. It is only limited by the signal-to-noise ratio (SNR) and the resolution achieved in the entrance slit and is, therefore, capable of yielding comprehensive measurement data for most instrument designs. In particular, it allows for a step-by-step tracing of the speckle statistics from the slit to the detector plane.

Based on this SFA measurement technique, we present a novel, stand-alone SFA prediction model, which solely relies on mathematical descriptions of the speckle statistic and its SFA impact. It includes the polarization effects of the diffuser, spatial and spectral averaging, and pixel averaging at the detector.

First, we review the definition of the spectral features amplitude (SFA) in Sect. 2. In Sect. 3, the revised SFA measurement technique used for our measurements is shown. We then present the stand-alone SFA prediction model in Sect. 4, which can be understood as a mathematical formulation of the SFA measurement technique. Finally, we compare the results of the prediction model to our measurement chain in detail in Sect. 5 to show its validity. In the last section, we discuss the applicability to a real instrument.

## 2   Spectral features amplitude

The term *spectral features amplitude* (SFA) was first proposed by van Brug and Courrèges-Lacoste (2007) as standardized way to quantify diffuser-induced wiggles in a spectrum measured by a space spectrometer instrument. They describe it as the magnitude of the features in a spectrum that are solely caused by the diffuser altering the solar reference spectrum. The SFA value is calculated as the standard deviation of the mean normalized detector signal over multiple spectral channels (see details in Sect. 3.1). The SFA value holds information about the amplitude of features. However, the data produced in this work, which are used to calculate the SFA, also allow for the estimation of the spectral extend of features. One usually may not draw conclusions about the absolute spectral positions of features with this approach. We will show that the instrument parameters used in this work lead to a spectral speckle extent smaller than the instrument detector pixel. This essentially allows for the treatment of the SFA as white noise at the detector level.

## 3   SFA measurement chain

In this section, the SFA measurement technique, introduced by Burns et al. (2017), is presented in a revised state. The goal of this technique is the reduction in experimental complexity and, therefore, systematic error contributions during data acquisition. First, the measurement principle is explained. Second, the used materials and the measurement procedures for the near infrared (NIR) and the short wavelength infrared (SWIR) channel are presented.

### 3.1   Principle

Figure 1 depicts the optical setup of an imaging spectrometer during solar calibration. The incoming sunlight is scattered by the diffuser. The scattering origin lies in the aperture plane with spatial coordinates $g$ and $h$, which is perpendicular to the light's direction of propagation. The angular field distribution at the aperture plane is imaged to the slit plane with the coordinates $x$ and $y$. The light is collimated onto a dispersive element (e.g., a diffraction grating), which

splits it into its spectral components. The spatial information in the $y$ direction of the slit is transformed into spectral information at the detector with coordinate $b$ by imaging the diffracted beams of different wavelengths onto a 2D array detector. Beams of the same wavelength (within the spectral resolution) are assigned the same spectral detector coordinate $b$. The spatial information in the $x$ direction is retained in the detector coordinate $a$. We relate the coordinates via the simplified linear spectrometer equations as follows: CE3

$$a = M_x x, \tag{1}$$

$$b = M_y y + k\lambda, \tag{2}$$

where $M_x$ and $M_y$ are the respective magnification factors in the $x$ and $y$ direction, $k = \mathrm{d}b/\mathrm{d}\lambda$ denotes the dispersion, and $\lambda$ is the wavelength. For these simplified equations to hold the magnification factors and the dispersion, they are assumed to be independent of the wavelength and the spatial position $(x, y)$. Also, the instrument point spread function (IPSF) is not accounted for. Sunlight is spatially coherent (Agarwal et al., 2004), and we can assume collimated light is illuminating the diffuser. Additionally, the sunlight's temporal coherence is on the order of femtoseconds (Hecht and Lippert, 2018) and much smaller than the typical detector integration time of several hundred milliseconds of an optical instrument. As a consequence, cross-coherence terms of interfering fields of different wavelengths average out, and the net intensity distributions at the slit and the detector planes are well approximated by the superposition of monochromatic intensities. The Sun disk is comprised of many incoherent point sources, which should be considered for angular averaging contributions and are not part of this work as they will only account for a single-point source. For the purpose of the SFA measurement, the sequence of optical components is subdivided into two parts. The first part, ranging from the illuminated diffuser through the telescope to the entrance slit, is represented by the optical setup in the lab. The second part comprises the rest of the optical system from the slit plane to the instrument-detector plane. The data acquired in the first part are used as input for a numerical simulation of the optical setup after the slit plane. The setup layout is shown in Fig. 3. The Sun is mimicked as a single field point with a tunable laser source, which is spectrally stabilized by a wavemeter, and it illuminates the diffuser through a linear polarizer at normal incident with respect to the diffuser plane. The distance between the single-mode fiber output and the diffuser is chosen such that the divergent beam illuminates the diffuser homogeneously over the size of the apertures. The second aperture blocks any unwanted angular contributions. A power meter placed next to the diffuser records a fixed fraction of the emitted laser power. The telescope images the scattered light onto a 2D array detector positioned in the focal plane. The focal plane of the telescope represents the slit plane in Fig. 1. The diffuser plane is tilted by 10° with respect to aperture and slit plane. This ensures that only scattered light contributes to the measurement. The telescope is aligned perpendicular to the aperture and slit plane.

For a measurement, monochromatic speckle intensities are recorded subsequently over a wavelength range $\lambda_1 \dots \lambda_N$ several times the spectral resolution $\lambda_{\mathrm{res}}$ of the real spectrometer that is being mimicked. An example of such an monochromatic speckle pattern is shown in Fig. 2a and d in the slit plane. The spectral tuning step size $\Delta\lambda$ in between images needs to be sufficiently small, in order to properly sample the change of the speckle patterns. The intermediate result is a 3D data set $I_{\mathrm{slit}}(x, y, \lambda)$ consisting of a spectrum of monochromatic speckle images, where $x$ and $y$ are the spatial coordinates in the slit plane and $\lambda$ is the wavelength. Every speckle image is mapped to a certain position $(a, b)$ at the focal plane, where all images are summed up in intensity. The summation on the intensity basis is justified as cross-coherence terms involving the interference of different wavelengths of actual sunlight, which will vanish for sufficiently long integration times. The summation procedure is detailed in Burns et al. (2017) and can be summarized as follows:

$$I_{\mathrm{det}}(a, b) = \frac{\Delta\lambda}{\lambda_{\mathrm{res}}} \sum_{\lambda=\lambda_1}^{\lambda_N} I_{\mathrm{slit}}\left(\frac{a}{M_x}, \frac{b - k\lambda}{M_y}, \lambda\right) \Theta\left(b - k\lambda\right), \tag{3}$$

where slit coordinates are expressed in terms of the detector coordinates using Eqs. (1) and (2) and the Heaviside function with $\Theta(y) = 0, y < 0$ and $\Theta(y) = 1, y \geq 0$. The result of the sum is a 2D intensity distribution in the focal plane of the instrument $I_{\mathrm{det}}(a, b)$, which is depicted in Fig. 2b and e. In a last step, $I_{\mathrm{det}}(a, b)$ is overlain with the actual instrument's detector pixel grid $(\tilde{a}, \tilde{b})$, and intensities belonging to the same pixel are summed, which are shown in Fig. 2c and f. The SFA is calculated as the standard deviation of the mean normalized detector pixel intensity distribution $I_{\mathrm{det,binned}}(\tilde{a}, \tilde{b})$.

## 3.2 Materials and procedure

Measurements are conducted in the NIR regime around 777 nm and in the SWIR regime around 1570 nm, which represent wavelength bands with commonly monitored data products, such as water vapor, clouds, $CO_2$, aerosols, or the $O_2$ absorption, which are commonly used to calculate the effective path length and the air mass factor (see Irizar et al., 2019, Meijer et al., 2019, or Voors et al., 2017). The experimental setup is shown in Sect. 3.1. As light sources, they serve tunable monochromatic external cavity diode lasers with single-mode output and an integrated optical isolator. They are stabilized via a proportional–integral–derivative (PID) loop with feedback data from a wavemeter, which uses a Fizeau interferometer. Connecting fibers between the laser, wavemeter, and output serve single-mode (SM) fibers since the spectral tuning range will be narrow. Also, SM fibers will introduce no additional speckle in contrast to multi-mode fibers. A linear polarizer ensures polarization stability. The round diffuser plate has a diameter of 70 mm and a thickness

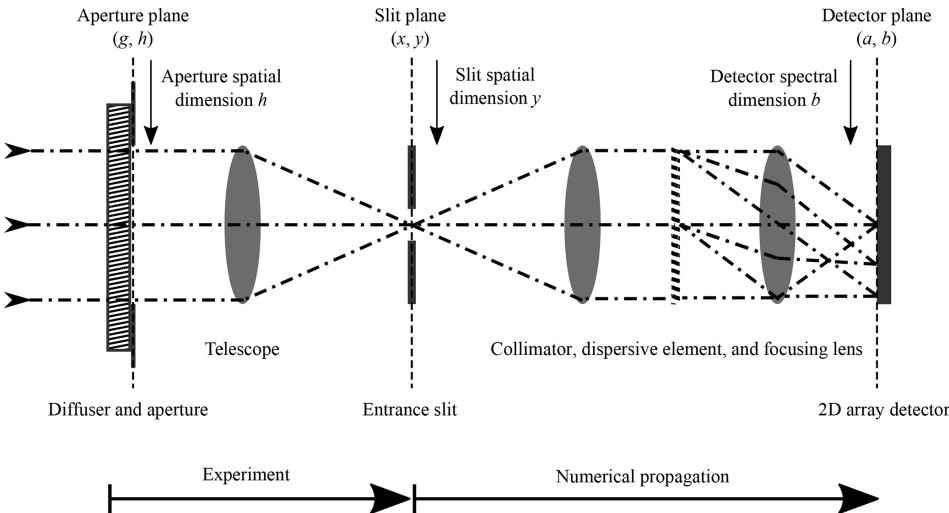

**Figure 1.** Optical setup of an imaging spectrometer during solar calibration. The sequence of optical components is subdivided into two parts. The first part is covered by the experimental setup in the lab, starting at the illuminated diffuser and ending at the slit in the telescope focal plane. The second part numerically propagates the images recorded in the slit plane to the instrument focal plane.

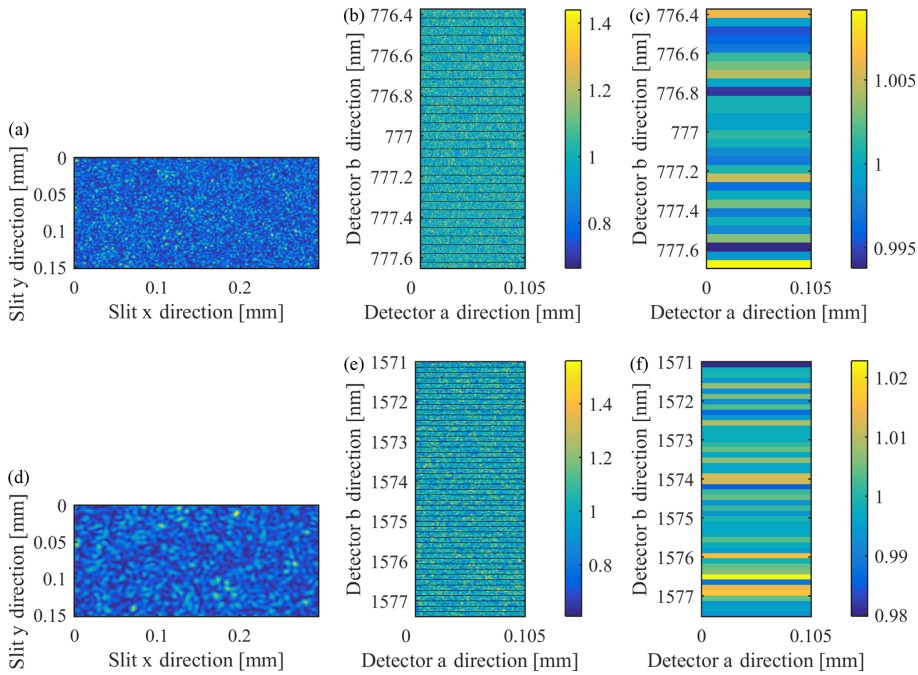

**Figure 2.** Speckle patterns in the NIR band **(a–c)** and SWIR band **(d–f)** at different stages in the measurement chain. Panels **(a)** and **(d)** are an example of a monochromatic speckle pattern in the slit plane; panels **(b)** and **(e)** are the speckle pattern integrated at the detector plane using Eq. (3) and normalized to its mean, where the horizontal lines denote the instrument detector pixel grid $(\tilde{a}, \tilde{b})$; and panels **(c)** and **(f)** are the final normalized integrated detector signal. The standard deviation taken over the pixel rows is the SFA.

of 3 mm. It is made of high-scattering fused silica HOD®-500 material featuring inhomogeneities of 20 μm TS3 or less. The spatial coherence length of the Sun's light is around 60 μm in the NIR and 120 μm in the SWIR band, according to Divitt and Novotny (2015). Hence, the material, which was selected for the Sentinel-5/UVNS according to Irizar

et al. (2019), is suited to the wavelengths in question and deemed a good choice for this study. The data collected with the power meter are used to normalize the acquired images. The round apertures are used to control the size of individual speckle correlation areas. The laser beam's uniformity at the diffuser plane was measured to be around 3 % in the NIR

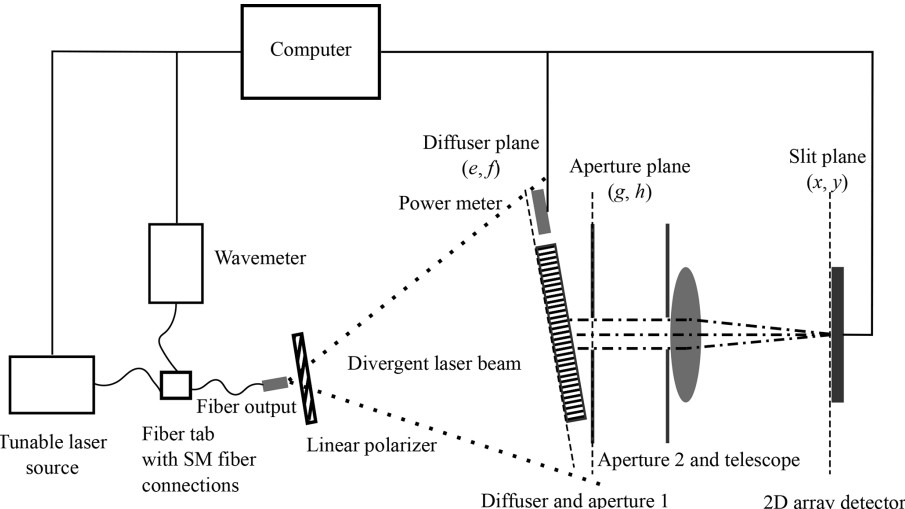

**Figure 3.** Layout of the experimental setup for measuring diffuser-induced monochromatic speckle patterns in the slit plane. Single-mode fibers are used to connect the laser source with the wavemeter and the output via a fiber tab and are indicated by curved lines.

band and 6 % in the SWIR band over the size of the aperture. There were no additional speckle contribution by the fibers detected. The telescope has a focal length of $f_{\text{tel}} = 1100\,\text{mm}$. For the NIR, the laser source has a center wavelength of 780 nm and a nominal linewidth of $6 \times 10^{-7}\,\text{nm}$. The Thorlabs 780HP SM fiber (SMF) type was used for all fiber connections in the NIR range. The charge-coupled device (CCD) detector features a 12.5 mm × 10.0 mm active area with 2750 × 2200 pixels of size 4.54 μm × 4.54 μm. The expected average 1D width of the speckle correlation area $S_{\text{d}}$ in the slit plane is calculated using the following:

$$S_{\text{d}} = \frac{2\lambda f_{\text{tel}}}{D\sqrt{\pi}}, \tag{4}$$

which was derived by Goodman (2007). The aperture's diameter is set to $D = 10\,\text{mm}$, and, thus, the speckles are sampled by 19 pixels in one dimension, which is deemed sufficiently fine to ensure that the sampling with the detector does not alter the speckle pattern significantly. The laser wavelength was tuned over the range of 776.4–777.7 nm, with a step size of $\Delta\lambda = 1\,\text{pm}$. The step size is chosen so that there is a non-zero correlation between subsequent speckle images. For the SWIR measurement, the laser source and the detector and the fiber splitter are replaced, and the Thorlabs SMF-28 was used for the fiber connections. The SWIR laser source center wavelength is 1550 nm, with a single-mode output of nominal $2 \times 10^{-6}\,\text{nm}$ linewidth. The detector is a 640 × 512 pixel InGaAs camera with a pixel size of 15.5 μm × 15.5 μm. The tuning range was 1571–1577.5 nm, with a step size of $\Delta\lambda = 3.1\,\text{pm}$. The aperture's diameter is set to $D = 13\,\text{mm}$, which means a sampling of 10 pixels per speckle in one dimension. This is smaller than for the NIR configuration but is a trade off between sampling and SNR.

## 4 Spectral features amplitude prediction model

The prediction model presented in the following is a mathematical formulation of the measurement method described in Sect. 3. It relies on the determination of the speckle statistics at different steps of the measurement chain. The relevant physical information about speckle averaging effects in the measurement chain lies in the intensity distributions. A single pattern $I$ is sampled by a finite but sufficient amount of pixels, so that the individual pixel size is small compared to the speckle size. The magnitude of the speckle effect in $I$ is described as the *speckle contrast* (Goodman, 2007, p. 28) as follows:

$$C = \frac{\sigma_I}{\langle I \rangle}, \tag{5}$$

where $\sigma_I$ is the standard deviation and $\langle I \rangle$ is the mean value of $I$ over all pixels. Under the general assumption that the individual statistics of the underlying fields are a circular complex Gaussian, a fully developed speckle pattern generated with linear polarized monochromatic light has a contrast of $C = 1$ (Goodman, 2007, p. 29). We adopt this assumption for this model. The speckle contrast is reduced by several averaging effects introduced by the spectrometer instrument. A reduction in $C$ is only achieved by the summation of intensity distributions showing a correlation smaller than unity. If the summation is on an amplitude basis (when the distributions can interfere), $C$ is not reduced (Goodman, 2007, Sect. 3.1.1). From this, it follows that only distributions which cannot interfere will impact $C$ and are therefore the subject of further discussions. Each one of the $N$ independent averaging effects is attributed to a certain amount of *degrees of freedom $M_n$* or effectively uncorrelated intensity distributions, which can be combined to a total averaging

factor $M$ (Goodman, 2007, p. 186) by the following:

$$M = \prod_n^N M_n. \tag{6}$$

The reduced speckle contrast will then be calculated as follows:

$$C_{\text{reduced}} = \frac{1}{\sqrt{M}} = \left( \sqrt{\prod_n^N M_n} \right)^{-1}. \tag{7}$$

In order to predict the contrast reduction, we identified $N = 3$ contributors, which can be assigned to different steps of the SFA measurement chain as follows:

1. generation of monochromatic diffuse depolarized light in the aperture plane $(g, h)$ yielding a factor $M_{\text{polarization}}$;

2. mapping intensities in the slit plane $(x, y)$ to instrument-detector positions $(a, b)$ contributing a factor $M_{\text{spectral}}$; and

3. integrating the instrument-detector pixels with a factor denoted by $M_{\text{detector}}$.

The predicted reduced speckle contrast at the instrument-detector plane, using Eq. (7), corresponds to the SFA and is as follows:

$$\text{SFA} = C_{\text{reduced}} = \frac{1}{\sqrt{M_{\text{polarization}} M_{\text{spectral}} M_{\text{detector}}}}. \tag{8}$$

## 4.1 Polarization averaging

The laser source emits a single polarization state, which is ensured with the polarizer. The diffused light leaving the volume diffuser can be treated as depolarized due to multi-scattering (Lorenzo, 2012, p. 85). This corresponds to two orthogonal polarization configurations or two effective intensity distributions which cannot interfere. Therefore step, $n = 1$ introduces two degrees of freedom $M_{\text{polarization}} = 2$ (Goodman, 2007, p. 49).

## 4.2 Spectral averaging

Step $n = 2$ leads to spectral averaging at the detector. We recall the finding from Sect. 3.1 that the net intensities in the field planes (slit and detector) can be treated as the superposition of monochromatic intensities for integration times much greater than the coherence time. Let us consider the acquired speckle intensities $I_n(x, y)$ and the underlying fields $\mathbf{A}_n(x, y)$ TS4, which are related by $I_n = |A_n|^2$. They are recorded at various wavelengths $\lambda_n$. The magnitude of the statistical change of subsequent speckle intensities $I_m$ and $I_n$, with a wavelength difference $\Delta\lambda_{nm} = |\lambda_n - \lambda_m|$, can be described in terms of the first-order field correlation coefficient $\mu_{mn}$, with the following:

$$\mu_{mn}(\lambda_n, \lambda_m) = \frac{\langle \mathbf{A}_m \mathbf{A}_n^* \rangle}{\sqrt{\langle I_m \rangle \langle I_n \rangle}}, \tag{9}$$

where $*$ denotes the complex conjugate. The field correlation is influenced by two effects which, in our case, are both frequency dependent. The first effect is due to changing light paths through the diffuser medium. The second effect takes into account the spatial offset $\Delta b = k\Delta\lambda_{nm}$ at the detector plane between individual speckle patterns $I_n$ induced by the dispersion (see Eq. 2). We start with the former contribution to the field correlation and follow the approach of Zhu et al. (1991), who presented an analytic equation for the angular correlation function of a slab geometry of a scattering media of thickness $d$, which can also be used for wavelength correlations as follows: TS5

$$F(\lambda_n, \lambda_m) =$$
$$\times \frac{\left[ \sinh\left(z_0\sqrt{q^2+\alpha^2}\right) + B\sqrt{q^2+\alpha^2}\cosh\left(z_0\sqrt{q^2+\alpha^2}\right) \right]^{(d+2B)/(z_0+B)}}{\left[1 + B^2(q^2+\alpha^2)\right]}$$
$$\times \sinh\left(d\sqrt{q^2+\alpha^2}\right) + 2B\sqrt{q^2+\alpha^2}\cosh\left(d\sqrt{q^2+\alpha^2}\right) \tag{10}$$

where we set $q = \sqrt{i6\pi \left| \frac{1}{\lambda_n} - \frac{1}{\lambda_m} \right| \beta n_s / l_t}$, with $n_s$ denoting the refractive index of the scattering material, $l_t$ being the transport mean free path used for anisotropic multi-scattering systems, and $\beta$ being a constant factor taking into account the contribution of the tilted diffuser plane $(e, f)$ with respect to the other planes and the specific geometry. The value $z_0$ describes the average penetration depth after which the light is scattered for the first time. It does not have great influences in transmission geometry and is approximated with the transport mean free path $l_t$. We set $\alpha = 0$, thereby ignoring absorption. The parameter for the boundary condition is given by $B = l_t \frac{2(1+R)}{3(1-R)}$, where $R$ is the reflection coefficient which is calculated using the Fresnel equations. It accounts for internal reflection due to the index of refraction mismatch at the boundaries.

The second contribution to the field correlation is due to changing spatial positions of speckle patterns which are distributed over the instrument detector in accordance with the spectral dispersion. This constitutes a spatial offset $\Delta b$ between the speckle intensities $I_n$ and $I_m$ at the detector plane $(a, b)$. Keeping in mind Eq. (1) and (2) for the transformation from the slit to the detector plane, the correlation of speckle fields, which are separated spatially by $\Delta b$, can be expressed as follows:

$$\Psi(\Delta a, \Delta b)|_{\Delta a=0} =$$
$$\frac{\int_{-\infty}^{\infty} |P(g,h)|^2 e^{-i\frac{2\pi}{\bar{\lambda}z}(g\Delta a + h\Delta b)} \mathrm{d}g\,\mathrm{d}h}{\int_{-\infty}^{\infty} |P(g,h)|^2 \mathrm{d}g\,\mathrm{d}h} \Bigg|_{\Delta a=0}, \tag{11}$$

where $P(g, h)$ is the aperture function of the imaging system and may be defined, e.g., for a circular aperture of diameter $D$, as $P(g,h) = \circ\left(\frac{2\sqrt{g^2+h^2}}{D}\right)$ TS6, which is 1 inside the aperture and 0 otherwise. $g$ and $h$ are the $a$- and $b$-coordinate

**Atmos. Meas. Tech., 14, 1–11, 2021**        **https://doi.org/10.5194/amt-14-1-2021**

representations in the aperture plane, $z$ is the distance between aperture and slit plane, and $\tilde{\lambda}$ is the mean wavelength (Goodman, 2007, p. 169). We set $\Delta a = 0$ since we are only interested in the offset in the spectral direction. Combining the two effects, we can model the correlation between the speckle fields as follows:

$$\mu_{mn}(\lambda_n, \lambda_m) = F(\lambda_n, \lambda_m)\Psi(\Delta b). \tag{12}$$

The accumulation of individual speckle patterns $I_n$ with field correlations $\mu_{mn}$ at the detector can be interpreted as the summation of partially correlated speckle intensities as follows:

$$I_{\text{det}}(a, b) = \sum_{n=1}^{N=\lambda_{\text{res}}/\Delta\lambda} I_n\left(\frac{a}{M_x}, \frac{b - k\lambda}{M_y}, \lambda_n\right). \tag{13}$$

The number of individual speckle intensities $I_n$ contributing to the sum at arbitrary detector coordinates $(a, b)$ is equal to the ratio of the spectral resolution $\lambda_{\text{res}}$ with the step size $\Delta\lambda$. This also applies to the mean intensities, as follows:

$$\langle I_{\text{det}}(a, b)\rangle = \sum_{n=1}^{N=\lambda_{\text{res}}/\Delta\lambda} \left\langle I_n\left(\frac{a}{M_x}, \frac{b - k\lambda}{M_y}, \lambda_n\right)\right\rangle. \tag{14}$$

Using an established method by Bevan (2009) and Goodman (2007), we define a coherency matrix with entries $J_{nm} = \langle \mathbf{A}_m \mathbf{A}_n^*\rangle$ and use Eq. (9) to obtain the following:

$$\mathbf{J} =$$
$$\begin{bmatrix} \langle I_1\rangle & \sqrt{\langle I_1\rangle\langle I_2\rangle}\mu_{1,2} & \cdots & \sqrt{\langle I_1\rangle\langle I_N\rangle}\mu_{1,N} \\ \sqrt{\langle I_1\rangle\langle I_2\rangle}\mu_{1,2}^* & \langle I_2\rangle & \cdots & \sqrt{\langle I_2\rangle\langle I_N\rangle}\mu_{2,N} \\ \vdots & \vdots & \ddots & \vdots \\ \sqrt{\langle I_1\rangle\langle I_N\rangle}\mu_{1,N}^* & \sqrt{\langle I_2\rangle\langle I_N\rangle}\mu_{2,N}^* & \cdots & \langle I_N\rangle \end{bmatrix}. \tag{15}$$

By the diagonalization of $\mathbf{J}$ with a unitary linear transformation $\mathbf{L}_0$, the ensemble of correlated speckle fields is transformed to a basis with no correlation between them.

$$\mathbf{J}' = \mathbf{L}_0\mathbf{J}\mathbf{L}_0^\dagger = \begin{bmatrix} \langle \tilde{I}_1\rangle & 0 & \cdots & 0 \\ 0 & \langle \tilde{I}_2\rangle & \cdots & 0 \\ \vdots & \vdots & \ddots & \vdots \\ 0 & 0 & \cdots & \langle \tilde{I}_N\rangle \end{bmatrix}, \tag{16}$$

where $\dagger$ denotes the Hermitian transpose operation. The total mean intensity $\langle I_{\text{det}}\rangle = \sum_n \langle I_n\rangle = \sum_n \langle \tilde{I}_n\rangle$ is conserved under this transformation but, in general, $\langle I_n\rangle \neq \langle \tilde{I}_n\rangle$. The complex coherence factor $\mu_{mn} = |\mu_{mn}|\exp(i\Phi_{nm})$ includes a phase $\Phi_{nm}$. However, due to the specific construction of $\mathbf{J}$, these phase terms can be omitted when calculating the eigenvalues (Dainty et al., 1975, Sect. 4.7.2). Finally, for the spectral degrees of freedom, we use the eigenvalues $\langle \tilde{I}_n\rangle$ of the coherency matrix to obtain the following:

$$M_{\text{spectral}} = \left(\frac{\langle I_{\text{det}}\rangle}{\sigma_{\text{det}}}\right)^2 = \frac{\left(\sum_n \langle \tilde{I}_n\rangle\right)^2}{\sum_n \langle \tilde{I}_n\rangle^2}. \tag{17}$$

Note that changing $\Delta\lambda$ and, thus, changing $N$ will not change the result of $M_{\text{spectral}}$ as long as $\Delta\lambda$ is sufficiently small to sample the covariance $\mu_{mn}$. The enabling property of the coherency matrix $\mathbf{J}$ is called Toeplitz, which implies an asymptotic behavior of its eigenvalues found by Grenander and Szegö (1958). Gray (2006) gives a simplified proof in corollary 2.1 and 2.2 that both the numerator and denominator in Eq. (17) converge for large $N$.

### 4.3 Detector averaging

In step $n = 3$, an averaging due to the integration of the instrument-detector pixel takes place. We already established that the resultant intensity distribution at the detector $I_{\text{det}}(a, b)$ is given by the summation in Eq. (13). This effect impacts the speckle contrast if individual speckles are not sufficiently oversampled by the instrument detector pixel grid $(\tilde{a}, \tilde{b})$. An analytical expression for the degrees of freedom $M_{\text{detector}}$ introduced by stationary speckles in one detector pixel with relative coordinates $(\Delta a, \Delta b)$ is given by the following:

$$M_{\text{detector}} =$$
$$\left[\frac{1}{A_{\text{D}}^2}\iint_{-\infty}^{\infty} K_{\text{D}}(\Delta a, \Delta b)|\mu_{\text{det}}(\Delta a, \Delta b)|^2 \text{d}\Delta a\,\text{d}\Delta b\right]^{-1}, \tag{18}$$

where $A_{\text{D}}$ is the area of a detector pixel, $K_{\text{D}}(\Delta a, \Delta b)$ is the autocorrelation function of the detector pixel, and $\mu_{\text{det}}(\Delta a, \Delta b)$ is the field correlation at the detector plane (Goodman, 2007, p. 108). In order to accurately describe $\mu_{\text{det}}$, one needs to account for the evolution of the speckle size during the summation in Eq. (13). Let us consider a single speckle correlation area $I_1(S_{\text{d}}/2 \leq |x - x_1|, S_{\text{d}}/2 \leq |y - y_1|, \lambda_1)$ with a spatial extent denoted by $S_{\text{d}}$ centered at $(x_1, y_1)$ in the slit. Its correlation relative to this position is described by the following:

$$\Psi(\Delta x, \Delta y) = \frac{\int_{-\infty}^{\infty}|P(g, h)|^2 e^{-i\frac{2\pi}{\tilde{\lambda}z}(g\Delta x + h\Delta y)}\text{d}g\,\text{d}h}{\int_{-\infty}^{\infty}|P(g, h)|^2\text{d}g\,\text{d}h}, \tag{19}$$

with $\Delta x = x - x_1$ and $\Delta y = y - y_1$ being relative coordinates. The function $F(\lambda_1, \lambda_n)$, introduced previously, characterizes how the correlation area develops after $n$ wavelength steps at the same position, $I_n(S_{\text{d}}/2 \leq |\Delta x|, S_{\text{d}}/2 \leq |\Delta y|, \lambda_n)$, with $n > 1$. In other words, it denotes the number of spectral steps after which a single speckle ceases to exist at a fixed position in the slit. The initial position of the speckle at the detector is $(a_1, b_1) = (M_x x_1, M_y y_1 + k\lambda_1)$. The subsequent contributions relative to the initial position are shifted in the $b$ direction by $k|\lambda_1 - \lambda_n|$ and have a magnitude denoted by $F(\lambda_1, \lambda_n)$. Therefore, the resultant speckle correlation function at the detector $|\mu_{\text{det}}(\Delta a, \Delta b)|^2$ is a convolution of $|\Psi(\Delta a, \Delta b)|^2$ with $|F(\lambda_n, \lambda_m)|^2$ as follows:

$$|\mu_{\text{det}}(\Delta a, \Delta b)|^2 = |\Psi(\Delta a, \Delta b)|^2 \circledast |F(\lambda_n, \lambda_m)|^2. \tag{20}$$

**Table 1.** Sample spectrometer parameters used for the measurement and prediction. They were chosen to represent a proposed instrument for the European Space Agency's CO2M mission (Meijer et al., 2019).

| Parameter | Value |
|---|---|
| Magnification $M_x$ | 0.34 |
| Magnification $M_y$ | 0.30 |
| Aperture diameter | 40.0 mm |
| Slit dimensions ($x$ and $y$ direction) | 295 and 152 μm |
| Detector dimensions ($a$ and $b$ direction) | 105 and 45 μm |
| Telescope focal length | 131 mm |
| Diffuser thickness $d$ | 3 mm |
| NIR specific | |
| Spectral resolution $\lambda_{\text{res}}$ | 0.128 nm |
| Spectral tuning range $\lambda_1 \ldots \lambda_N$ | 776.4–777.7 nm |
| Tuning step size $\Delta\lambda$ | 1 pm |
| Refractive index of diffuser material $n_s(\lambda)$ | 1.454 |
| Mean free path $l_t(\lambda)$ | $(59.3 \pm 0.4)$ μm |
| SWIR specific | |
| Spectral resolution $\lambda_{\text{res}}$ | 0.4 nm |
| Spectral tuning range $\lambda_1 \ldots \lambda_N$ | 1571–1577.5 nm |
| Tuning step size $\Delta\lambda$ | 3.1 pm |
| Refractive index of diffuser material $n_s(\lambda)$ | 1.444 |
| Mean free path $l_t(\lambda)$ | $(67.8 \pm 0.5)$ μm |

The symbol $\circledast$ denotes the convolution.

## 5  Results and discussion

In the following, we present and compare the SFA results from the measurement chain of Sect. 3 with the ones from the prediction model of Sect. 4 in the NIR and SWIR regime. $M_{\text{spectral}}$ can be interpreted as the average number of speckle patterns generated by the diffuser per spectral channel at the given wavelengths. $M_{\text{detector}}$ can be understood as the average number of speckle correlation areas influencing the measurement in a detector pixel. The values of relevant parameters used are given in Table 1. They were chosen to represent a proposed instrument for the European Space Agency's (ESA) CO2M mission (Meijer et al., 2019). In Fig. 4, the measured Pearson correlations between speckle patterns $I_n(\lambda_n)$ and $I_m(\lambda_m)$, with respect to their relative spectral distance, are shown as blue stars for the NIR band at 776 nm and the SWIR band at 1571 nm, respectively. All wavelength shift combinations up to 0.1 nm are averaged for 120 images. Error bars are omitted, because the standard error of the mean is too small to be displayed. The red line denotes a fit to Eq. (10), leading to the values for $l_t$ given in Table 1. They are in agreement with the supplier's given value of $l_t = 56$ μm at 500 nm wavelength.

Table 2 shows the SFA values of the measurements and predictions with their corresponding intermediate averaging factors $M_{\text{polarization}}$, $M_{\text{spectral}}$, and $M_{\text{detector}}$, as introduced in Sect. 4. Their counterparts from the measurement are deducted by calculating the speckle contrast at intermediate steps in the measurement chain. To verify the factor $M_{\text{polarization}} = 2$, a linear polarizer is placed after the diffuser, and the measured speckle contrast, compared to the nominal case, rises by a factor of $\sqrt{2}$. Additionally, the polarization axis is rotated to different random positions without changing the result, which confirms the assumption made in Sect. 4 that the light exiting the diffuser is depolarized. With an ideal measurement chain the speckle contrast expected in the slit plane for monochromatic polarized speckles is $C_{\text{slit,ideal}} = 1$. This is the numerator in Eq. (8). The measured contrast in the slit is smaller, probably due to detector noise, as suggested by Postnov et al. (2019). Webster et al. (2003) attributed the reduced measured contrast to straylight from multiple reflections in their setup. The experimental factors $M_{\text{spectral,measured}}$ and $M_{\text{detector,measured}}$ are calculated in similar way, using the following relations:

$$C_{\text{spectral,measured}} = \frac{\langle C_{\text{slit,measured}} \rangle}{\sqrt{M_{\text{spectral,measured}}}}, \tag{21}$$

$$C_{\text{detector,measured}} = \frac{C_{\text{spectral,measured}}}{\sqrt{M_{\text{detector,measured}}}}. \tag{22}$$

The results show a good agreement between the prediction model and measurement and are well within the estimated error margins, which are given in the $1\sigma$ interval. While the spectral extent of a speckle correlation area at the detector can be derived from the width of $\mu_{\text{det}}$, one can also see from the values of $M_{\text{detector}}$ that the extent must be small compared to a pixel. This allows for the treatment of the SFA as white noise on a detector level, which can be a goal of the instrument design. The uncertainties $\Delta M_{\text{spectral,measured}}$ and $\Delta M_{\text{detector,measured}}$ are estimated by considering fluctuations in the two contributors of $F$ and $\Psi$ to the field correlation $\mu_{mn}$ in Eq. (12) and $\mu_{\text{det}}$ in Eq. (20). For $F$, the measured standard deviation of each data point in Fig. 4 is taken as a measure, which amounts to 1 %–2 % in the NIR and 2 %–3 % in the SWIR band. The average fluctuations of $\Psi$ were determined by measuring its width over 120 speckle patterns with autocorrelation to be 1.3 % and 2.7 % in the NIR and SWIR, respectively. The impact on the $M_{\text{spectral,measured}}$ and $M_{\text{detector,measured}}$ is estimated by a Monte Carlo propagation of both uncertainties. It yields that those two factors are the major contributors to the uncertainty $\Delta M_{\text{spectral}}$. This approach also accounts for detector noise, if it caused the fluctuations. The impact on $M_{\text{detector}}$, however, is small. It was deduced from Ahn and Fessler (2003), that $\Delta M_{\text{detector}}$ is primarily caused by the uncertainty of the standard deviation estimator for a small number of samples. In our case, the sample number equals the number of detector pixels (see Fig. 2), which is $n = 30$ for the NIR and $n = 48$ for the SWIR mea-

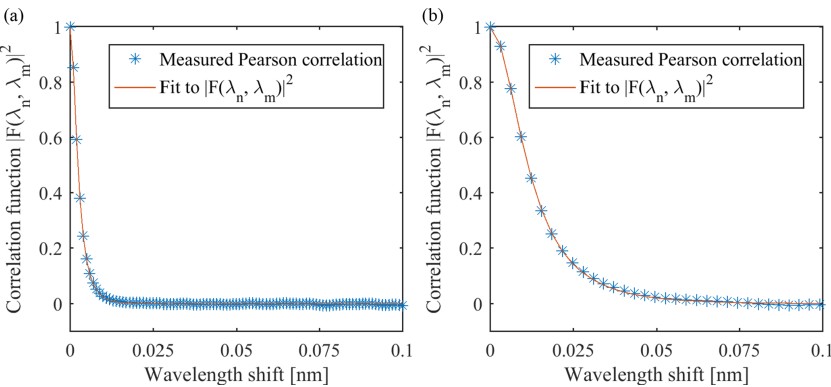

**Figure 4.** Measurement of the correlation function $|F(\lambda_n, \lambda_m)|^2$ in **(a)** the NIR (776 nm) and **(b)** the SWIR band (1571 nm). Blue stars denote the measured Pearson correlations between speckle patterns $I_n(\lambda_n)$ and $I_m(\lambda_m)$. All wavelength shift combinations up to 0.1 nm are averaged for 120 images. Error bars are omitted because the standard error of the mean value is too small to be displayed. The red graph denotes the fit of the measured data points to Eq. (10), with **(a)** $l_t = (59.3 \pm 0.4)\,\mu m$ and **(b)** $l_t = (67.8 \pm 0.5)\,\mu m$.

**Table 2.** Comparison SFA results of the measurement chain with the prediction model. Measurement uncertainties are given in the $1\sigma$ interval.

| Type | $M_{polarization}$ | $M_{spectral}$ | $M_{detector}$ | SFA (%) |
|------|------|------|------|------|
| Measurement NIR | 2 | $55.9 \pm 0.7$ | $(6.1 \pm 1.8) \times 10^2$ | $0.38 \pm 0.06$ |
| Prediction NIR | 2 | 56.5 | $5.7 \times 10^2$ | 0.39 |
| Measurement SWIR | 2 | $29.9 \pm 0.8$ | $(1.7 \pm 0.4) \times 10^2$ | $0.99 \pm 0.12$ |
| Prediction SWIR | 2 | 30.0 | $1.8 \times 10^2$ | 0.96 |

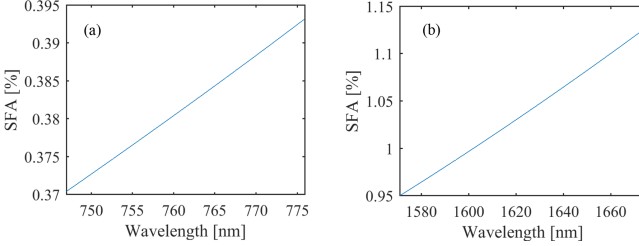

**Figure 5.** Scaling of the SFA with the wavelength of a CO2I-like instrument in the **(a)** NIR and **(b)** SWIR band, using the prediction model.

surement. The uncertainties presented also reflect the fact that the SFA is not constant over a wavelength range of a few nanometers, as illustrated in Fig. 5. It shows plots of the SFA scaling, with the wavelength of a CO2I-like instrument in the NIR and SWIR band, calculated with the prediction model. The linear dependency only holds under the assumption of a constant dispersion, which is usually not the case for a realistic instrument. However, this result gives a clear indication of the general scaling of the speckle effect.

## 6 Conclusions

We demonstrated a comprehensive and numerical approach to quantify diffuser-induced spectral features during the solar calibration of space imaging spectrometers, which is based on established speckle theory concepts. We compared our prediction results with our current measuring method and observed a good agreement. We also gave an indication regarding the wavelength dependency of the effect. The presented speckle averaging mechanisms are not a complete representation of the real in-orbit situation of an instrument. The effect of the Sun's disk, which consists of many incoherent point sources distributed over a 0.5° angle, needs to be taken into account together with the averaging due to the movement of the instrument relative to the Sun. Also, unlike the used laser point sources for the measurements, the Sun's light features an additional orthogonal polarization state, adding two polarization configurations to $M_{polarization}$ in the case of a high-scattering volume diffuser. The presented approach can be used for other diffuser types and optical geometries as well. It provides a solid starting point for future investigations into angular averaging mechanisms, which will complement the description of speckle reduction effects in imaging spectrometers of this type.

*Data availability.* The data sets generated and/or analyzed for this work are available from the corresponding author on reasonable request, subject to the confirmation of Airbus Defence and Space GmbH.

*Author contributions.* FR was responsible for the acquisition and the analysis of the measurement data and was supported by all co-authors. FR developed the prediction model, which was supported by JC, who provided insights about polarization contributions, spectral averaging, and the properties of coherency matrices, and revised by CK and MW. FR prepared the paper, with contributions and critical revision from all co-authors. TS7

*Competing interests.* The authors declare that they have no conflict of interest.

*Acknowledgements.* The fused silica diffuser HOD®-500 used in this work was provided by Frank Nürnberg and Bernhard Franz, Heraeus Conamic, Germany.

*Review statement.* This paper was edited by Saulius Nevas and reviewed by Peter Gege and one anonymous referee.

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

## Remarks from the language copy-editor

CE1    Please note the slight adjustments to this section.

CE2    It is our standard to use italics sparingly, and thus, they have been removed from the paper after the first instance. Should you feel very strongly that italics are necessary in some instances, please let me know, and we can try to find a solution.

CE3    Please note that I have added the construction "as follows" or "in the following" or variations thereof before each of the equations in this paper (where applicable). This is in alignment with our house standard. However, if you wish to rework the phrase from "as follows" to read "in the following" or something similar in some instances in order to reflect a more accurate meaning of your text, please let me know, and I will gladly accommodate you.

## Remarks from the typesetter

TS1    Please provide previous affiliation.

TS2    The composition of Figs. 1 and 3 has been adjusted to our standards. This also includes language adjustments to Figs. 1–3.

TS3    Please confirm.

TS4    Please make sure all vectors are identified in bold italic font and all matrices in bold roman font.

TS5    Please confirm line break in Eq. 10.

TS6    Please confirm ∘.

TS7    Please also mention JK and DW in this section. It should be clear who contributed to which part of the paper.

TS8    Please provide event, date and venue.

TS9    Please provide more information.

TS10    Please provide date and venue of the conference.

TS11    Please provide event, date and venue.

TS12    Please provide date of last access.

TS13    Please provide place of publication.

TS14    Please provide more information.

TS15    Please provide place of publication.

TS16    Please provide event, date and venue.

TS17    Please provide event, date and venue.

TS18    Please provide date of last access.

TS19    Please provide date and venue of the symposium.

TS20    Please provide place of publication.

TS21    Please provide event, date and venue.

TS22    Please provide event, date and venue.

TS23    Please provide page range or article number.

TS24    Please provide event, date and venue.

TS25    Please provide event, date and venue.

TS26    Please provide event, date and venue.

TS27    Please provide event, date and venue.

TS28    Please provide date and venue of the conference.

TS29    Please provide page range or article number.