# Peer review of "Prediction Model for Diffuser Induced Spectral Features in Imaging Spectrometers"

_Atmospheric Measurement Techniques, 2020_

## Referee Comment (RC1) · Anonymous Referee #1 · 29 Sep 2020

**General comments**

This is a well written paper. However, what this paper is currently missing in my opinion is visualization of the measured and processed data. Can you please add visualizations of actual measured data (both spectral and images) and results from each of the modeling steps? It would be interesting for readers to see how speckle patterns of that diffuser look and the pattern propagation to the final imaging plane. Can you also add some figures from the laser spectra used? Just as background information you could include in the introduction that sources like lasers produce temporal speckle and diffusers produce spatial speckle. Speckle pattern created by a diffuser can be averaged for example by rotating the diffuser during the measurement.

[Figure]

This is slightly off from the focus on algorithm development itself, but very important what comes to the proper operation of the instrument. What are the criteria for the diffuser to reduce speckle effect and how does that affect overall performance of the instrument? Are there any tradeoffs?

Maybe you could add reasoning why this specific glass volume diffuser was selected and refer to studies on some diffuser contamination and radiation tests? Contamination/degradation of the diffusers is to my understanding a major issue in satellite measurements. This is at least a problem at UV and visible and it has much larger effect than speckle. Since diffraction limit increases at longer wavelengths, at NIR and SWIR diffuser speckle is worse than at visible spectral range. You could mention this in the paper.

**Specific comments:**
**Page 1, row 22:** "... spectrometers with fine spectral resolution and strict demands to radiometric accuracy ..." You could specify these "strict demands" in the paper. At least stray light is usually a problem in imaging spectrometers.

**Page 2, row 29:** "... end-to-end measurements by van Brug and Courrèges-Lacoste (2007) as well as models for different speckle averaging effects ..." Can you explain in detail what end-to-end measurements were these and what existing speckle averaging methods are available (both hardware and software)?

**Page 5, Figure 2:** Regarding the setup, please specify the type of optical fibers used between the tunable laser source and the fiber tab and between the fiber tab and the fiber output. You could say that since your spectral tuning range is narrow, you can use single mode fibers to transmit the laser beam and create uniform illumination.

You could also show in a figure how spatially uniform the radiation output from the single mode fiber is before it hits the diffuser. Is there any spatial speckle created by the single mode fiber? You could mention that multimode fibers should not be used as they generate severe spatial speckle that can be worse than the diffuser speckle. The speckle pattern by the multimode fiber changes when the fiber bends only slightly. In addition, can you please draw Figure 2 so that it is easier to see which cables are optical fibers and which of them are electrical cables.

**Page 5, Figure 2:** Please replace "Powermeter" with "Power meter"

**Page 5, rows 109–111:** Can you please give references to these data products?

**Page 5, row 110:** Please replace "... CO2, Aerosols, or the O2 absorption ..." with "... $CO_2$, aerosols, or the $O_2$ absorption ..."

**Page 6, rows 117–118:** "For the NIR the laser source has a center wavelength of 780 nm and a nominal linewidth of 300 kHz." Can you give the nominal linewidth in wavelengths?

**Page 6, rows 125–126:** "The SWIR laser source center wavelength is 1550 nm, with single mode output of nominal 150 kHz linewidth." Can you give the nominal linewidth in wavelengths?

**Page 6, row 129:** Please define a speckle oversampling ratio.

**Page 7, row 162:** Please define $f_m$.

**Page 7, row 167:** You have an error in the spatial offset equation $\Delta b = k \frac{c}{\Delta f}$. Because $\lambda = \frac{c}{f}$ and its derivative is $\Delta \lambda = -\frac{c}{f^2} \Delta f$, the spatial offset equation should be $\Delta b = k \Delta \lambda = -k \frac{c}{f^2} \Delta f$. How does correcting this affect the results?

**Page 8, row 172:** Please present exact equation for the path length probability density function $p_l(l)$.

**Page 8, row 178:** There is one extra parenthesis, please replace $\Delta b(\Delta f))$ with $\Delta b(\Delta f)$.

**Page 8, row 180:** There is a typing error in Equation (10), the integrals' limits should be $\int_{-\infty}^{\infty}$ (instead of $\int_{\infty}^{\infty}$ ).

**Page 8, row 181:** "... $P(h)$ is the aperture function of the imaging system ..." Please present the exact equation for $P(h)$.

**Page 8, row 180:** There is a typing error in Equation (18), the integrals' limits should be $\int_{-\infty}^{\infty}$ (instead of $\int_{\infty}^{\infty}$).

**Page 10, row 225:** "Therefore, the resultant speckle correlation function at the detector $\mu_{det}(\Delta a, \Delta b)$ is a convolution of ...". Please present $\mu_{det}(\Delta a, \Delta b)$ as an equation.

**Page 10, row 250:** "We assume, that detector noise is averaged in this step" Can you please add what the noise properties of the detector are (e.g. in V/$\sqrt{\text{Hz}}$) and what integration times were used?

**Page 10, Subsection 4.4 Predicted SFA:** Since this subsection includes only one sentence, to me it makes more sense to remove this subsection 4.4 and move the equation to the beginning of Section 4, right after Eq. (7). There you could also define $M_{polarization}$, $M_{spectral}$, and $M_{detector}$ and link these symbols to the steps 1, 2, and 3.

**Page 11, Table 2:** Based on the values of $M_{polarization}$, $M_{spectral}$, and $M_{detector}$, the SFA on the first row of Table 2 should be 0.0039 (not 0.0040). Can you please give SFAs in percents in Table 2?

**Page 11, row 254:** "It is also dependent on detector noise, which explains slightly higher averaging factors than predicted." Can you predict noise properties of the detector used and add them to the calculations?

**In results section:** How much diffuser speckle contributes to the overall measurement uncertainty?

---

## Referee Comment (RC2) · Peter Gege (Referee) · 24 Nov 2020

General comments

During the operation of spectrally highly resolving satellite sensors (SCIAMACHY, OMI) small wiggles were observed in the spectrometer data whose origin was unclear in the beginning. Finally they could be explained by interference patterns (speckles) of the diffuser used for radiometric in-orbit calibration. The manuscript describes a model to predict these features.

The developed model is physically sound and has been validated using laboratory measurements at two wavelengths. While the technical content is complete and well explained, I miss some motivation and explanation of the effect in the Introduction and

an illustration and discussion of the practical relevance in the Results and Discussion section. I recommend publication after these minor revisions are made.

Specific comments

The modeled effect is not well known in the scientific community. It is difficult to believe that sunlight induces noticeable interference effect in solids because speckles are typical for lasers, but not for natural light. You should mention in the Introduction the puzzling spectral wiggles in the order of a few percent discovered in data of the SCIA-MACHY and OMI instruments (described and illustrated by van Brug et al., 2004). Interference effects require that the coherence length is larger than the size of material inhomogeneities, thus you should quantify the coherence length of sunlight and compare it with the typical scale of inhomogeneities in diffuser material. According to Divitt and Novotny (2015) the coherence length of sunlight is 80 x wavelength, which corresponds to 62 $\mu$m and 126 $\mu$m at the wavelengths of your measurements (777 nm, 1570 nm). You should also mention that the effect is not restricted to diffusers, but occurs for all static measurements of a solid. Point out its relevance for laboratory calibration of spectrometers and spectral measurements using a fixed set-up of target, spectrometer and light source; and explain that the effect vanishes if one of these is moved or tilted during the measurement, as in case of remote sensing.

The importance of the effect and the model for practical applications should be graphically illustrated in the Results and Discussion sections. In particular a plot of the wavelength dependency should be added due to its high relevance for spectral measurements. The wavelength dependency cannot be assessed from the equations, thus a graphical illustration would help the reader to grasp the importance of the described effect. Furthermore, such a plot would allow comparing at least qualitatively the modelled spectral dependency with observed spectral patterns, e.g. as in Fig. 1 of van Brug et al. (2004).

Line 21f: "Since Spectral Features are of statistical nature and cannot be mitigated by

any post-processing steps". Correction may indeed be difficult, but not because the effect is of statistical nature, but because the intensity of the pattern and its position on the focal plane are difficult to calculate accurately because they depend on a number of parameters which are difficult to measure with sufficient accuracy (temperature, pressure, isotropy of incident light field).

Technical corrections

line 18: "diffuser introduces a statistical interference phenomenon" I suggest to delete statistical (as it is a geometric effect, not a statistical one) and replace "phenomenon" by "pattern"

Line 50: unwanted "features". Unclear: What do you mean with unwanted? Unexplained?

Line 51f: "The SFA value is then calculated as the standard deviation of the normalized signal over a certain spectral width, that includes multiple features." The definition of SFA is unclear: normalized to what? What means "certain spectral width"? Is standard deviation calculated over time or over wavelength? The definition only gets clear at line 106f together with Eq. (3). Improve here the explanation and refer for details to Eq. (3).

Line 76f: "Sun's light . . . is assumed to be spatially coherent giving the distance from the Sun to the Earth and the limited acceptance angle of the spectrometer." The coherence of light from a spatially incoherent spherical source is in fact valid immediately beyond a distance of a few wavelengths (Agarwal et al. 2004), thus the spatial coherence of sunlight has nothing to do with the Earth-sun distance. Quantify instead the coherence length, e.g. by citing the result 80 * Lambda by Divitt and Novotny (2015). Also the influence of the acceptance angle of the spectrometer is not clear. Either explain or remove it.

Line 77f: "the temporal coherence is very short compared to the detector integration time, which is in the order of seconds" Quantify the temporal coherence and replace

seconds by milli-seconds. The sunlight coherence time is 3 fs according to Herman et al. (2014) who cite the "Optics" book of Hecht (2016).

Line 120: add a reference for Eq. (4).

Line 233f: "the fitted mean free path length of our diffuser sample is determined as ls = $53\mu$m" Explain how you fitted the free path length. The mean free path length depends on wavelength as the scattering probability is wavelength-dependent. Hence, add the wavelength.

Line 300: Coernicus –> Copernicus

References

G. S. Agarwal, G. Gbur, and E. Wolf, "Coherence properties of sunlight," Opt. Lett. 29, 459461 (2004)

S. Divitt and L. Novotny, "Spatial coherence of sunlight and its implications for light management in photovoltaics,"Optica 2, 95103 (2015)

A. Herman, M. Sarrazin, and O. Deparis, "The fundamental problem of treating light incoherence in photovoltaicsand its practical consequences," New J. Phys. 16, 013022 (2014).

E. Hecht, Optics (Pearson Higher Education, Harlow, England, 2016), 5th ed.

van Brug, H. H., Vink, R., Schaarsberg, J. G., Courreges-Lacoste, G. B., and Snijders, B.: Speckles and their effects in spectrometers due to on-board diffusers, in: Earth Observing Systems IX, edited by Barnes, W. L. and Butler, J. J., SPIE, https://doi.org/10.1117/12.559596, 2004.

---

## Author Comment (AC1) · 8 Dec 2020

**Author response to referee comment #1**

December 8, 2020

We thank referee #1 for the time spent reading the manuscript and the productive and helpful comments. We have addressed the referee's comments on a point to point basis as below for consideration. All page and line numbers refer to the first version of the manuscript.

**1 General comments**

R1: *This is a well written paper. However, what this paper is currently missing in my opinion is visualization of the measured and processed data. Can you please add visualizations of actual measured data (both spectral and images) and results from each of the modeling steps? It would be interesting for readers to see how speckle patterns of that diffuser look and the pattern propagation to the final imaging plane. Can you also add some figures from the laser spectra used? Just as background information you could include in the introduction that sources like lasers produce temporal speckle and diffusers produce spatial speckle. Speckle pattern created by a diffuser can be averaged for example by rotating the diffuser during the measurement.*

Response: We will add figures illustrating the speckle patterns in the slit and detector plane. The width of the laser line is presented in Section 3. We will also add more detailed explanations and motivations regarding speckle effects and some mitigation principles.

R1: *This is slightly off from the focus on algorithm development itself, but very important what comes to the proper operation of the instrument. What are the criteria for the diffuser to reduce speckle effect and how does that affect overall performance of the instrument? Are there any tradeoffs?*

Response: Generally, the SFA will be lower the more independent speckle patterns per wavelength band a diffuser generates. For example, the NIR instrument used in this work would generate about 56 independent patterns ($M_{spectral}$) over a spectral range equal to the resolution $\lambda_{res}$ at 776nm. This number is influenced by the sensitivity of the diffuser with respect to wavelength change. This sensitivity is significantly higher for transmission geometries, since

it will yield a wider range of possible optical path differences. This effect also scales with the thickness of the diffuser. As far as our experience goes: a thicker diffuser also means less transmission and therefore less signal on the detector during calibration.

R1: *Maybe you could add reasoning why this specific glass volume diffuser was selected and refer to studies on some diffuser contamination and radiation tests? Contamination/degradation of the diffusers is to my understanding a major issue in satellite measurements. This is at least a problem at UV and visible and it has much larger effect than speckle. Since diffraction limit increases at longer wavelengths, at NIR and SWIR diffuser speckle is worse than at visible spectral range. You could mention this in the paper.*

Response: The specific diffuser material was chosen for the measurements since it was qualified for the Sentinel 5/UVNS (see Irizar et al. (2019)). We will mention this when describing the material in section 3.2. The impact of speckles generated by longer wavelengths can be seen in the smaller averaging factor $M_{spectral}$ of the SWIR band compared to the NIR. We will point this out during the discussion of the results.

**2 Specific comments**

**Page1, row 22:** R1: *"... spectrometers with fine spectral resolution and strict demands to radiometric accuracy..." You could specify these "strict demands" in the paper. At least stray light is usually a problem in imaging spectrometers.*

Response: We will clarify that diffuser speckles need to be considered as part of the radiometric accuracy error budget next to other contributors such as straylight or polarization.

**Page 1, row 29:** R1: *"...end-to-end measurements by van Brug and Courrèges-Lacoste (2007) as well as models for different speckle averaging effects..." Can you explain in detail what end-to-end measurements were these and what existing speckle averaging methods are available (both hardware and software)?*

Response: We will add an explanation regarding the end-to-end measurements. The authors are convinced, that the effects of diffuser speckle can not be reliably characterized with representative end-to-end setups (which includes a full spectrometer, telescope optics, diffuser and light source). The suppression of the speckle effects is implicit to the design of the instrument. Therefore, if one were to measure speckle residuals with a certain setup, it can not be representative for an instrument that is supposed to have a neglectable residual speckle amplitude (SFA).

**Page 5, Figure 2:** R1: *Regarding the setup, please specify the type of optical*

*fibers used between the tunable laser source and the fiber tab and between the fiber tab and the fiber output. You could say that since your spectral tuning range is narrow, you can use single mode fibers to transmit the laser beam and create uniform illumination. You could also show in a figure how spatially uniform the radiation output from the single mode fiber is before it hits the diffuser. Is there any spatial speckle created by the single mode fiber? You could mention that multimode fibers should not be used as they generate severe spatial speckle that can be worse than the diffuser speckle. The speckle pattern by the multimode fiber changes when the fiber bends only slightly. In addition, can you please draw Figure 2 so that it is easier to see which cables are optical fibers and which of them are electrical cables.*

Response: We will specify the exact fiber types used, include the result of spatial beam uniformity measurements at the diffuser plane, and add the referee's suggested comments regarding multi mode fibers. We will also correct Figure 2. The spatial uniformity measurements showed no apparent spatial speckles generated by the fibers, which we will mention as well.

**Page 5, Figure 2:** R1: *Please replace "Powermeter" with "Power meter".*

Response: Done.

**Page 5, rows 109-111:** R1: *Can you please give references to these data products?*

Response: Done.

**Page 5, row 110:** R1: *Please replace "... CO2, Aerosols, or the O2 absorption..." with "... $CO_2$, aerosols, or the $O_2$ absorption ..."*

Response: Done.

**Page 6, rows 117-118:** R1: *"For the NIR the laser source has a center wavelength of 780 nm and a nominal linewidth of 300 kHz." Can you give the nominal linewidth in wavelengths?*

Response: Done.

**Page 6, rows 125-126:** R1: *"The SWIR laser source center wavelength is 1550 nm, with single mode output of nominal 150 kHz linewidth." Can you give the nominal linewidth in wavelengths?*

Response: Done.

**Page 6, row 129:** R1: *Please define a speckle oversampling ratio*

Response: We will provide an explanation without this confusing term. It states how many detector pixel the speckle correlation areas are being sampled with.

**Page 6, row 162:** R1: *Please define $f_m$.*
Response: We will change this part slightly to be more understandable, also introducing the symbol $\Delta f$ correctly.

**Page 7, row 167:** R1: *You have an error in the spatial offset equation.... How does correcting this affect the results?*

Response: The symbol "$\Delta f$" was used incorrect at some instances, which led to the misleading expression for $\Delta b$. It will be corrected accordingly.

**Page 8, row 172:** R1: *Please present exact equation for the path length probability density function $p(l)$.*

Response: We will use a direct analytic expression for $F(\Delta f)$ given by Zhu et al. (1991), which lead to similar results. However, this way one does not need to detour of calculating $p(l)$ first and taking the Fourier transform after. Also this expression gives direct access to the angular correlation function, too, which is interesting for estimations of angular contributions in the future. For the interested reader see Patterson et al. (1989), where an expression for $p(l)$ is derived explicitly.

**Page 8, row 178:** R1: *There is one extra parenthesis, please replace $\Delta b(\Delta f))$ with $\Delta b(\Delta f)$.*

Response: We will use the symbol "$\Delta b$" without the explicit frequency dependence.

**Page 8, row 180:** R1: *There is a typing error in Equation (10), the integrals' limits should be $\int_{-\infty}^{\infty}$ instead of $\int_{\infty}^{\infty}$.*

Response: Done.

**Page 8, row 181:** R1: *"...$P(h)$ is the aperture function of the imaging system ...". Please present the exact equation for $P(h)$.*

Response: We will give an example for a circular aperture now.

**Page 10, row 225:** R1: *"Therefore, the resultant speckle correlation function at the detector $\mu_{det}(\Delta a, \Delta b)$ is a convolution of ...". Please present $\mu_{det}(\Delta a, \Delta b)$ as an equation.*

Response: We will present an equation.

**Page 10, row 250:** R1: *"We assume, that detector noise is averaged in this step" Can you please add what the noise properties of the detector are (e.g. in $V/\sqrt{Hz}$) and what integration times were used?*

Response: We will present a complete rework regarding the uncertainties for the averaging factors and the SFA.

**Page 10, Subsection 4.4 Predicted SFA:** R1: *Since this subsection includes only one sentence, to me it makes more sense to remove this subsection 4.4 and move the equation to the beginning of Section 4, right after Eq. (7). There you could also define $M_{polarization}$, $M_{spectral}$, and $M_{detector}$ and link these symbols to the steps 1, 2, and 3.*

Response: Done.

**Page 11, Table 2:** R1: *Based on the values of $M_{polarization}$, $M_{spectral}$, and $M_{detector}$, the SFA on the first row of Table 2 should be 0.0039 (not 0.0040). Can you please give SFAs in percents in Table 2?*

Response: We will include an error analysis and also fixed a minor inconsistency in our spectrometer propagation, which caused some systematic deviations in the measured averaging factors, especially for the NIR band.

**Page 11, row 254:** R1: *"It is also dependent on detector noise, which explains slightly higher averaging factors than predicted." Can you predict noise properties of the detector used and add them to the calculations?*

Response: We will rework the discussion of uncertainties.

**Results section:** R1: *How much diffuser speckle contributes to the overall measurement uncertainty?*

Response: The relative spectral radiometric accuracy (RSRA) budget of ESA's CO2M mission is 0.5% for all bands (see Meijer et al.). However, the SFA results presented in this work should not be directly compared to this budget, since it does not account for angular averaging effects. The SFA including those effects can be two orders of magnitude lower.

**3   References**

Zhu, J. X., Pine, D. J., and Weitz, D. A.: Internal reflection of diffusive light in
random media, Phys. Rev. A, 44, 3948–3959, https://doi.org/10.1103/PhysRevA.44.3948,
https://link.aps.org/doi/10.1103/PhysRevA.44.3948, 1991.
Patterson, M. S., Chance, B., andWilson, B. C.: Time resolved reflectance
and transmittance for the noninvasive measurement of tissue optical properties,
Appl. Opt., 28, 2331–2336, https://doi.org/10.1364/AO.28.002331, http://ao.osa.org/abstract.cfm?URI=a
28-12-2331, 1989.
Irizar, J., Melf, M., Bartsch, P., Koehler, J., Weiss, S., Greinacher, R., Erdmann,
M., Kirschner, V., Albinana, A. P., and Martin, D.: Sentinel- 5/UVNS, in: In-
ternational Conference on Space Optics 2018, edited by Sodnik, Z., Karafolas,
N., and Cugny, B., vol. 11180, pp. 41 – 58, 315 International Society for Optics
and Photonics, SPIE, https://doi.org/10.1117/12.2535923, https://doi.org/10.1117/12.2535923,
2019.
Meijer, Y., Boesch, H., Bombelli, A., Brunner, D., Buchwitz, M., Ciais, P.,
Crisp, D., Engelen, R., Holmlund, K., Houweling, S., Janssens- Meanhout,
G., Marshall, J., Nakajima, M., B.Pinty, Scholze, M., Bezy, J.-L., Drinkwater,
M., Fehr, T., Fernandez, V., Loescher, A., Nett, H., and Sierk, B.: Coerni-
cus CO2 Monitoring Mission Requirements Document, techreport 2, European
Space Agency, Earth and Mission Science Division, 2019.

---

## Author Comment (AC2) · 8 Dec 2020

**Author response to referee comment #2**

December 4, 2020

We thank referee #2 for the time spent reading the manuscript and the productive and helpful comments. We have addressed the referee's comments on a point to point basis as below for consideration. All page and line numbers refer to the first version of the manuscript.

**1 General comments**

R2: The developed model is physically sound and has been validated using laboratory measurements at two wavelengths. While the technical content is complete and well explained, I miss some motivation and explanation of the effect in the Introduction and an illustration and discussion of the practical relevance in the Results and Discussion section. I recommend publication after these minor revisions are made.

Response: We will add more detailed explanations and motivations regarding speckle effect into the introduction and extend the discussion in order to better illustrate our results.

**2 Specific comments**

R2: The modeled effect is not well known in the scientific community. It is difficult to believe that sunlight induces noticeable interference effect in solids because speckles are typical for lasers, but not for natural light. You should mention in the Introduction the puzzling spectral wiggles in the order of a few percent discovered in data of the SCIAMACHY and OMI instruments (described and illustrated by van Brug et al., 2004).

Response: We will clarify these aspects in the introduction.

R2: Interference effects require that the coherence length is larger than the size of material inhomogeneities, thus you should quantify the coherence length of sunlight and compare it with the typical scale of inhomogeneities in diffuser material. According to Divitt and Novotny (2015) the coherence length of sunlight

is 80 x wavelength, which corresponds to 62  $\mu$ m and 126  $\mu$ m at the wavelengths of your measurements (777 nm, 1570 nm).

Response: According to the specifications given by the manufacturer the scattering centers of the diffuser material have a maximum diameter of 20 microns. We will add a comparison in Section 3.2.

R2: The importance of the effect and the model for practical applications should be graphically illustrated in the Results and Discussion sections. In particular a plot of the wavelength dependency should be added due to its high relevance for spectral measurements. The wavelength dependency cannot be assessed from the equations, thus a graphical illustration would help the reader to grasp the importance of the described effect. Furthermore, such a plot would allow comparing at least qualitatively the modelled spectral dependency with observed spectral patterns, e.g. as in Fig. 1 of van Brug et al. (2004).

Response: We will add plots of the wavelength dependence for both bands in the discussion.

R2: You should also mention that the effect is not restricted to diffusers, but occurs for all static measurements of a solid. Point out its relevance for laboratory calibration of spectrometers and spectral measurements using a fixed set-up of target, spectrometer and light source; and explain that the effect vanishes if one of these is moved or tilted during the measurement, as in case of remote sensing.

Response: We will introduce the reader more broadly into the subject of speckle and some mitigation methodes in the introduction. However, we would like to point out, that the movement or the tilting of parts in the optical system are usually not implemented in space applications, since they would involve additional moving parts, which is usually to be avoided. The only change of the geometry would be the angle of incident due to the movement of the instrument relativ to the sun. Angular averaging effects, however, are not part of this work.

Line 21f: R2: "Since Spectral Features are of statistical nature and cannot be mitigated by any post-processing steps". Correction may indeed be difficult, but not because the effect is of statistical nature, but because the intensity of the pattern and its position on the focal plane are difficult to calculate accurately because they depend on a number of parameters which are difficult to measure with sufficient accuracy (temperature, pressure, isotropy of incident light field).

Response: We will adapt this passage regarding the term "statistical".

**3 Technical corrections**

**Line 18:** R2: : "diffuser introduces a statistical interference phenomenon" I suggest to delete statistical (as it is a geometric effect, not a statistical one) and replace "phenomenon" by "pattern".

Response: We see the argument of the Referee here and will follow his suggestion to remove "statistical" for this specific formulation. However, we still see the need for a quasi statistical treatment of the effect, because the resulting intensity of the speckle pattern seen by the detector is essentially unpredictable.

**Line 50:** R2: : unwanted "features". Unclear: What do you mean with unwanted? Unexplained?.

Response: What is meant here, are the "features", that are caused solely by the diffuser. They are "unwanted" in the sense, that they alter the solar reference spectrum, which is recorded during calibration. We will change this sentence to clarify this.

**Line 51f:** R2: : The SFA value is then calculated as the standard deviation of the normalized signal over a certain spectral width, that includes multiple features." The definition of SFA is unclear: normalized to what? What means "certain spectral width"? Is standard deviation calculated over time or over wavelength? The definition only gets clear at line 106f together with Eq. (3). Improve here the explanation and refer for details to Eq. (3).

Response: We will replace "certain wavelength range" by "multiple spectral channels". The standard deviation is taken over the normalized detector signal, which essentially is calculating it over wavelength. We will clarify this.

Line 76f: R2: : "Sun's light... is assumed to be spatially coherent giving the distance from the Sun to the Earth and the limited acceptance angle of the spectrometer." The coherence of light from a spatially incoherent spherical source is in fact valid immediately beyond a distance of a few wavelengths (Agarwal et al. 2004), thus the spatial coherence of sunlight has nothing to do with the Earth-sun distance. Quantify instead the coherence length, e.g. by citing the result 80 \* Lambda by Divitt and Novotny (2015). Also the influence of the acceptance angle of the spectrometer is not clear. Either explain or remove it.

Response: We wanted to point out that the spatially coherent Sun light incident on the diffuser can be treated as collimated under the mentioned conditions, which is also matched by our experimental setup. We will rework this part.

Line 77f: R2: : "the temporal coherence is very short compared to the detector integration time, which is in the order of seconds" Quantify the temporal coherence and replace seconds by milli-seconds. The sunlight coherence time is 3 fs according to Herman et al. (2014) who cite the "Optics" book of Hecht (2016).

Response: We will rework this part.

Line 120: R2: : add a reference for Eq. (4).

Response: Done.

**Line 233f:** R2: : "the fitted mean free path length of our diffuser sample is determined as  $ls = 53\mu m$ ". Explain how you fitted the free path length. The mean free path length depends on wavelength as the scattering probability is wavelength-dependent. Hence, add the wavelength.

Response: We will use different values for  $l_s$  for each band. We will add an explanation on how this values were obtained. The reference value of  $l_s = 56 \mu m$  given by the manufacturer is for  $\lambda = 500 nm$ . We obtained more realistic values for  $l_s$  at the employed wavelengths using the approach by Zhu et al. (1991) of calculating the frequency correlation function  $F(\Delta f)$ .

**Line 300:** R2: : Coernicus  $\rightarrow$  Copernicus.

Response: Done.

**4 References**

Zhu, J. X., Pine, D. J., and Weitz, D. A.: Internal reflection of diffusive light in random media, Phys. Rev. A, 44, 3948–3959, https://doi.org/10.1103/PhysRevA.44.3948, https://link.aps.org/doi/10.1103/PhysRevA.44.3948, 1991.

---

## Author Response (AR1)

**Author response to the revised manuscript version**

This document repeats the questions of all referees with the corresponding authors' responses on a point to point basis. Additionally, it indicates for every comment the changes made in manuscript and explains the reasoning of the authors if needed. All page and line references refer to the revised manuscript version. Appended to this document is a differential view between the first and revised manuscript version for convenient tracking of the applied changes.

**1 General comments Referee 1**

R1: *This is a well written paper. However, what this paper is currently missing in my opinion is visualization of the measured and processed data. Can you please add visualizations of actual measured data (both spectral and images) and results from each of the modeling steps? It would be interesting for readers to see how speckle patterns of that diffuser look and the pattern propagation to the final imaging plane. Can you also add some figures from the laser spectra used? Just as background information you could include in the introduction that sources like lasers produce temporal speckle and diffusers produce spatial speckle. Speckle pattern created by a diffuser can be averaged for example by rotating the diffuser during the measurement.*

Response: We will add figures illustrating the speckle patterns in the slit and detector plane. The width of the laser line is presented in Section 3. We will also add more detailed explanations and motivations regarding speckle effects and some mitigation principles.

Changes in manuscript: We added Figure 2, which shows the measurement data at the intermediate steps in the measurement chain. The width of the laser line is given in lines 138 and 148 according to supplier specifications. We introduced into the subject of speckle and some reduction aspects in the introduction (lines 20-34).

R1: *This is slightly off from the focus on algorithm development itself, but very important what comes to the proper operation of the instrument. What are the criteria for the diffuser to reduce speckle effect and how does that affect overall performance of the instrument? Are there any tradeoffs?*

Response: Generally, the SFA will be lower the more independent speckle patterns per wavelength band a diffuser generates. For example, the NIR instrument used in this work would generate about 56 independent patterns ($M_{spectral}$) over a spectral range equal to the resolution $\lambda_{res}$ at 776nm. This number is influenced by the sensitivity of the diffuser with respect to wavelength change. This sensitivity is significantly higher for transmission geometries, since it will yield a wider range of possible optical path differences. This effect also scales with the thickness of the diffuser. As far as our experience goes: a thicker diffuser also means less transmission and therefore less signal on the detector during calibration.

Changes in manuscript: No specific changes were made.

R1: *Maybe you could add reasoning why this specific glass volume diffuser was selected and refer to studies on some diffuser contamination and radiation tests? Contamination/degradation of the diffusers is to my understanding a major issue in satellite measurements. This is at least a problem at UV and visible and it has much larger effect than speckle. Since diffraction limit increases at longer wavelengths, at NIR and SWIR diffuser speckle is worse than at visible spectral range. You could mention this in the paper.*

Response: The specific diffuser material was chosen for the measurements since it was qualified for the Sentinel 5/UVNS (see Irizar et al. (2019)). We will mention this when describing the material in section 3.2. The impact of speckles generated by longer wavelengths can be seen in the smaller averaging factor $M_{spectral}$ of the SWIR band compared to the NIR. We will point this out during the discussion of the results.

Changes in manuscript: A justification for the diffuser material is now given in lines 132-134. An interpretation to the spectral averaging factor $M_{spectral}$ is given in lines 262/263 and Figure 5 shows the SFA scaling with wavelength explicitly.

**2  Specific comments Referee 1**

**Page1, row 22:** R1: *"... spectrometers with fine spectral resolution and strict demands to radiometric accuracy..." You could specify these "strict demands" in the paper. At least stray light is usually a problem in imaging spectrometers.*

Response: We will clarify that diffuser speckles need to be considered as part of the radiometric accuracy error budget next to other contributors such as straylight or polarization.

Changes in manuscript: Lines 24-26.

**Page 1, row 29:** R1: *"...end-to-end measurements by van Brug and Courrèges-Lacoste (2007) as well as models for different speckle averaging effects..." Can you explain in detail what end-to-end measurements were these and what existing speckle averaging methods are available (both hardware and software)?*

Response: We will add an explanation regarding the end-to-end measurements. The authors are convinced, that the effects of diffuser speckle can not be reliably characterized with representative end-to-end setups (which includes a full spectrometer, telescope optics, diffuser and light source). The suppression of the speckle effects is implicit to the design of the instrument. Therefore, if one

were to measure speckle residuals with a certain setup, it can not be representative for an instrument that is supposed to have a neglectable residual speckle amplitude (SFA).

Changes in manuscript: Detailed explanation regarding those aspects can be found in lines 20-34 and 38-44.

**Page 5, Figure 2:** R1: *Regarding the setup, please specify the type of optical fibers used between the tunable laser source and the fiber tab and between the fiber tab and the fiber output. You could say that since your spectral tuning range is narrow, you can use single mode fibers to transmit the laser beam and create uniform illumination. You could also show in a figure how spatially uniform the radiation output from the single mode fiber is before it hits the diffuser. Is there any spatial speckle created by the single mode fiber? You could mention that multimode fibers should not be used as they generate severe spatial speckle that can be worse than the diffuser speckle. The speckle pattern by the multimode fiber changes when the fiber bends only slightly. In addition, can you please draw Figure 2 so that it is easier to see which cables are optical fibers and which of them are electrical cables.*

Response: We will specify the exact fiber types used, include the result of spatial beam uniformity measurements at the diffuser plane, and add the referee's suggested comments regarding multi mode fibers. We will also correct Figure 2. The spatial uniformity measurements showed no apparent spatial speckles generated by the fibers, which we will mention as well.

Changes in manuscript: Information about the used fibers were added in lines 127-129, 138/139, and 147. Spatial uniformity is given in lines 135-137. Figure 2 is now Figure 3 and is corrected.

**Page 5, Figure 2:** R1: *Please replace "Powermeter" with "Power meter".*

Response: Done.

Changes in manuscript: see Figure 3.

**Page 5, rows 109-111:** R1: *Can you please give references to these data products?*

Response: Done.

Changes in manuscript: see lines 124-125.

**Page 5, row 110:** R1: *Please replace "... CO2, Aerosols, or the O2 absorption..." with "... $CO_2$, aerosols, or the $O_2$ absorption ..."*

Response: Done.

Changes in manuscript: line 123.

**Page 6, rows 117-118:** R1: *"For the NIR the laser source has a center wavelength of 780 nm and a nominal linewidth of 300 kHz." Can you give the nominal linewidth in wavelengths?*

Response: Done.

Changes in manuscript: see line 138.

**Page 6, rows 125-126:** R1: *"The SWIR laser source center wavelength is 1550 nm, with single mode output of nominal 150 kHz linewidth." Can you give the nominal linewidth in wavelengths?*

Response: Done.

Changes in manuscript: see line 148.

**Page 6, row 129:** R1: *Please define a speckle oversampling ratio*

Response: We will provide an explanation without this confusing term. It states how many detector pixel the speckle correlation areas are being sampled with.

Changes in manuscript: The sampling of speckle in the slit plane are explained in lines 143 and 151.

**Page 6, row 162:** R1: *Please define $f_m$.*
Response: We will change this part slightly to be more understandable, also introducing the symbol $\Delta f$ correctly.

Changes in manuscript: We now use symbols with wavelength dependency instead of frequency, because the description is easier to understand in our opinion.

**Page 7, row 167:** R1: *You have an error in the spatial offset equation.... How does correcting this affect the results?*

Response: The symbol "$\Delta f$" was used incorrect at some instances, which led to the misleading expression for $\Delta b$. It will be corrected accordingly.

Changes in manuscript: We now give symbols with wavelength dependency instead of frequency, which simplifies this expression.

**Page 8, row 172:** R1: *Please present exact equation for the path length probability density function $p(l)$.*

Response: We will use a direct analytic expression for $F(\Delta f)$ given by Zhu et al. (1991), which lead to similar results. However, this way one does not need to detour of calculating $p(l)$ first and taking the Fourier transform after. Also this expression gives direct access to the angular correlation function, too, which is interesting for estimations of angular contributions in the future. For the interested reader see Patterson et al. (1989), where an expression for $p(l)$ is derived explicitly.

Changes in manuscript: see Equation (10) and following lines.

**Page 8, row 178:** R1: *There is one extra parenthesis, please replace $\Delta b(\Delta f))$ with $\Delta b(\Delta f)$.*

Response: We will use the symbol "$\Delta b$" without the explicit frequency dependence.

Changes in manuscript: $\Delta b$ is now defined in line 194.

**Page 8, row 180:** R1: *There is a typing error in Equation (10), the integrals' limits should be $\int_{-\infty}^{\infty}$ instead of $\int_{\infty}^{\infty}$.*

Response: Done.

Changes in manuscript: see Equation (11).

**Page 8, row 181:** R1: *"...$P(h)$ is the aperture function of the imaging system ...". Please present the exact equation for $P(h)$.*

Response: We will give an example for a circular aperture now.

Changes in manuscript: see line 212.

**Page 10, row 225:** R1: *"Therefore, the resultant speckle correlation function at the detector $\mu_{det}(\Delta a, \Delta b)$ is a convolution of ...". Please present $\mu_{det}(\Delta a, \Delta b)$ as an equation.*

Response: We will present an equation.

Changes in manuscript: see Equation 20.

**Page 10, row 250:** R1: *"We assume, that detector noise is averaged in this step" Can you please add what the noise properties of the detector are (e.g. in $V/\sqrt{Hz}$) and what integration times were used?*

Response: We will present a complete rework regarding the uncertainties for the averaging factors and the SFA.

Changes in manuscript: First, a discussion of detector noise is given in line 278-279 suggesting why speckles are not measured with contrast of unity, which also happened in other studies. However, this contribution is constant and does not change the measurement result, because we compensate it by comparing contrast levels and then assume an initial contrast of unity. Secondly, any detector noise is indirectly accounted for in the estimation of uncertainties for the averaging factors discussed in lines 286-297. We basically characterized fluctuations in the two functions $F$ and $\Psi$, that basically govern both averaging factors. Additionally, a more significant factor for $M_{detector}$ seems to be a statistical effect regarding the standard deviation estimator. A direct approach is tedious in our opinion, since noise should vary significantly over a single speckle image.

**Page 10, Subsection 4.4 Predicted SFA:** R1: *Since this subsection includes only one sentence, to me it makes more sense to remove this subsection 4.4 and move the equation to the beginning of Section 4, right after Eq. (7). There you could also define $M_{polarization}$, $M_{spectral}$, and $M_{detector}$ and link these symbols to the steps 1, 2, and 3.*

Response: Done.

Changes in manuscript: see lines 173-178.

**Page 11, Table 2:** R1: *Based on the values of $M_{polarization}$, $M_{spectral}$, and $M_{detector}$, the SFA on the first row of Table 2 should be 0.0039 (not 0.0040). Can you please give SFAs in percents in Table 2?*

Response: We will include an error analysis and also fixed a minor inconsistency in our spectrometer propagation, which caused some systematic deviations in the measured averaging factors, especially for the NIR band.

Changes in manuscript: SFA values is given in percent now in Table 2. For the derivation of uncertainties see lines 286-297. We fixed a numerical (rounding) problem in our spectrometer propagation. This involved slightly changing the spectral resolution of the instrument in the NIR. We also had to shorten the tuning range because of some systematic environmental influences. For the SWIR band we slightly increased the step size $\Delta\lambda$ from 2.5 pm to 3.1 pm (see Table 1) and were able to increase the tuning range in an attempt to decrease the uncertainty for $M_{detector}$ (which depends on number of spectral pixel rows, hence the tuning range). Also we increased the apertures' diameter from 10 mm to 13 mm to increase SNR.

**Page 11, row 254:** R1: *"It is also dependent on detector noise, which explains slightly higher averaging factors than predicted." Can you predict noise proper-*

*ties of the detector used and add them to the calculations?*

Response: We will rework the discussion of uncertainties.

Changes in manuscript: see changes of the comment above.

**Results section:** R1: *How much diffuser speckle contributes to the overall measurement uncertainty?*

Response: The relative spectral radiometric accuracy (RSRA) budget of ESA's CO2M mission is 0.5% for all bands (see Meijer et al.). However, the SFA results presented in this work should not be directly compared to this budget, since it does not account for angular averaging effects. The SFA including those effects can be two orders of magnitude lower.

Changes in manuscript: Since we did not present a detailed analysis of angular averaging mechanisms in this study a absolute comparison to a realistic error budgets seems inappropriate at this point. However, we do explain in the conclusion how the results in this work fit into a complete description of all averaging effects in an imaging spectrometer that are present to our knowledge.

**3 General comments Referee 2**

R2: *The developed model is physically sound and has been validated using laboratory measurements at two wavelengths. While the technical content is complete and well explained, I miss some motivation and explanation of the effect in the Introduction and an illustration and discussion of the practical relevance in the Results and Discussion section. I recommend publication after these minor revisions are made.*

Response: We will add more detailed explanations and motivations regarding speckle effect into the introduction and extend the discussion in order to better illustrate our results.

Changes in manuscript: We added a more comprehensive introductory part in lines 20-44. We give an interpretation of the determined averaging factor in lines 262-264 and the scaling with wavelength in Figure 5 and lines 296-299.

**4 Specific comments Referee 2**

R2: *The modeled effect is not well known in the scientific community. It is difficult to believe that sunlight induces noticeable interference effect in solids because speckles are typical for lasers, but not for natural light. You should mention in the Introduction the puzzling spectral wiggles in the order of a few percent discovered in data of the SCIAMACHY and OMI instruments (described and illustrated by van Brug et al., 2004).*

Response: We will clarify these aspects in the introduction.

Changes in manuscript: see lines 20f.

R2: *Interference effects require that the coherence length is larger than the size of material inhomogeneities, thus you should quantify the coherence length of sunlight and compare it with the typical scale of inhomogeneities in diffuser material. According to Divitt and Novotny (2015) the coherence length of sunlight is 80 x wavelength, which corresponds to 62 µm and 126 µm at the wavelengths of your measurements (777 nm, 1570 nm).*

Response: According to the specifications given by the manufacturer the scattering centers of the diffuser material have a maximum diameter of 20 microns. We will add a comparison in Section 3.2.

Changes in manuscript: see lines 130-134.

R2: *The importance of the effect and the model for practical applications should*

*be graphically illustrated in the Results and Discussion sections. In particular a plot of the wavelength dependency should be added due to its high relevance for spectral measurements. The wavelength dependency cannot be assessed from the equations, thus a graphical illustration would help the reader to grasp the importance of the described effect. Furthermore, such a plot would allow comparing at least qualitatively the modelled spectral dependency with observed spectral patterns, e.g. as in Fig. 1 of van Brug et al. (2004).*

Response: We will add plots of the wavelength dependence for both bands in the discussion.

Changes in manuscript: see Figure 5 and lines 296-299 as well as Figure 4.

R2: *You should also mention that the effect is not restricted to diffusers, but occurs for all static measurements of a solid. Point out its relevance for laboratory calibration of spectrometers and spectral measurements using a fixed set-up of target, spectrometer and light source; and explain that the effect vanishes if one of these is moved or tilted during the measurement, as in case of remote sensing.*

Response: We will introduce the reader more broadly into the subject of speckle and some mitigation methodes in the introduction. However, we would like to point out, that the movement or the tilting of parts in the optical system are usually not implemented in space applications, since they would involve additional moving parts, which is usually to be avoided. The only change of the geometry would be the angle of incident due to the movement of the instrument relativ to the sun. Angular averaging effects, however, are not part of this work.

Changes in manuscript: See lines 20-34.

**Line 21f:** R2: *"Since Spectral Features are of statistical nature and cannot be mitigated by any post-processing steps". Correction may indeed be difficult, but not because the effect is of statistical nature, but because the intensity of the pattern and its position on the focal plane are difficult to calculate accurately because they depend on a number of parameters which are difficult to measure with sufficient accuracy (temperature, pressure, isotropy of incident light field).*

Response: We will adapt this passage regarding the term "statistical".

Changes in manuscript: We removed the term statistical (line 18) and build the argumentation beginning in line 20 to line 32 regarding the quasi statistical treatment of the speckle effect.

**5 Technical corrections Referee 2**

**Line 18:** R2: *: "diffuser introduces a statistical interference phenomenon" I suggest to delete statistical (as it is a geometric effect, not a statistical one) and replace "phenomenon" by "pattern".*

Response: We see the argument of the Referee here and will follow his suggestion to remove "statistical" for this specific formulation. However, we still see the need for a quasi statistical treatment of the effect, because the resulting intensity of the speckle pattern seen by the detector is essentially unpredictable.

Changes in manuscript: see line 18 and lines 20-32.

**Line 50:** R2: *: unwanted "features". Unclear: What do you mean with unwanted? Unexplained?.*

Response: What is meant here, are the "features", that are caused solely by the diffuser. They are "unwanted" in the sense, that they alter the solar reference spectrum, which is recorded during calibration. We will change this sentence to clarify this.

Changes in manuscript: see lines 61f.

**Line 51f:** R2: *: The SFA value is then calculated as the standard deviation of the normalized signal over a certain spectral width, that includes multiple features." The definition of SFA is unclear: normalized to what? What means "certain spectral width"? Is standard deviation calculated over time or over wavelength? The definition only gets clear at line 106f together with Eq. (3). Improve here the explanation and refer for details to Eq. (3).*

Response: We will replace "certain wavelength range" by "multiple spectral channels". The standard deviation is taken over the normalized detector signal, which essentially is calculating it over wavelength. We will clarify this.

Changes in manuscript: see lines 62f. We refer to Section 3.2 for details.

**Line 76f:** R2: *: "Sun's light... is assumed to be spatially coherent giving the distance from the Sun to the Earth and the limited acceptance angle of the spectrometer." The coherence of light from a spatially incoherent spherical source is in fact valid immediately beyond a distance of a few wavelengths (Agarwal et al. 2004), thus the spatial coherence of sunlight has nothing to do with the Earth-sun distance. Quantify instead the coherence length, e.g. by citing the result 80 * Lambda by Divitt and Novotny (2015). Also the influence of the acceptance angle of the spectrometer is not clear. Either explain or remove it.*

Response: We wanted to point out that the spatially coherent Sun light incident on the diffuser can be treated as collimated under the mentioned conditions, which is also matched by our experimental setup. We will rework this part.

Changes in manuscript: see lines 88-89.

**Line 77f:** R2: : "the temporal coherence is very short compared to the detector integration time, which is in the order of seconds" Quantify the temporal coherence and replace seconds by milli-seconds. The sunlight coherence time is 3 fs according to Herman et al. (2014) who cite the "Optics" book of Hecht (2016).

Response: We will rework this part.

Changes in manuscript: see lines 87-92.

**Line 120:** R2: : add a reference for Eq. (4).

Response: Done.

Changes in manuscript: see line 143.

**Line 233f:** R2: : "the fitted mean free path length of our diffuser sample is determined as ls = 53μm". Explain how you fitted the free path length. The mean free path length depends on wavelength as the scattering probability is wavelength-dependent. Hence, add the wavelength.

Response: We will use different values for $l_s$ for each band. We will add an explanation on how this values were obtained. The reference value of $l_s = 56\mu m$ given by the manufacturer is for $\lambda = 500nm$. We obtained more realistic values for $l_s$ at the employed wavelengths using the approach by Zhu et al. (1991) of calculating the frequency correlation function $F(\Delta f)$.

Changes in manuscript: see Equation (10) and following lines, Table 1, Figure 4, and lines 265-270. Note that we used the wavelength dependence for $F$ and that we changed $l_s$ to $l_t$, which denotes the transport mean free path. This is more appropriate for anisotropic multi scattering systems.

**Line 300:** R2: : Coernicus → Copernicus.

Response: Done.

Changes in manuscript: see line 362.

[revised manuscript text omitted]